# Microarchitecture is Destiny: Performance and Accuracy of Quantized LLMs on Consumer Hardware

## Abstract

While the deployment of out-of-the-box quantization on consumer-grade hardware is widespread, its impact on Large Language Models (LLMs) reveals a complex, twofold phenomenon that questions the prevailing assumption. This study presents a rigorous empirical evaluation across four generations of NVIDIA GPUs, uncovering two core, often counter-intuitive, findings. First, contrary to the prevailing view that quantization universally degrades performance on complex tasks, the analysis demonstrates that for large models (14B+ parameters), popular 8-bit and 4-bit quantization schemes can yield substantial accuracy improvements on mathematical reasoning benchmarks compared to their 16-bit floating-point counterparts, which suffer from system-level bottlenecks in resource-constrained environments. Second, the investigation reveals that for smaller models prone to overfitting, the noise introduced by these same quantization schemes can act as an effective computational regularizer, unexpectedly enhancing generalization. The performance analysis further establishes that once VRAM capacity is met, the GPU microarchitecture's support for low-precision integer arithmetic, rather than VRAM size, becomes the primary determinant of inference throughput. These findings provide a more nuanced perspective that moves beyond a simplistic trade-off, offering practitioners an evidence-based framework for navigating the interplay between model scale, hardware capabilities, and reasoning fidelity.

## 1 Introduction

The proliferation of powerful open-source Large Language Models (LLMs) like Qwen2.5 and DeepSeek-R1, coupled with user-friendly frameworks such as Ollama, has democratized access to advanced AI. This accessibility, however, masks a critical knowledge gap: many practitioners deploy quantized models on consumer-grade hardware without a deep understanding of the complex interplay between model architecture, quantization artifacts, and the underlying hardware. This often leads to suboptimal outcomes, from unexpected abrupt performance degradation to silent, catastrophic failures in model accuracy.

This study confronts this gap through a systematic, empirical evaluation. The analysis makes a clear distinction between the roles of VRAM and microarchitecture in determining performance. While VRAM capacity acts as a binary gatekeeper—determining whether a model can be loaded at all—the evidence indicates that once this prerequisite is met, the GPU's microarchitecture becomes the primary determinant of inference speed. Specifically, hardware support for low-precision integer arithmetic (e.g., via Tensor Cores) constitutes the key driver of throughput for quantized models.

The investigation, conducted across four generations of NVIDIA GPUs, reveals a more nuanced, twofold phenomenon. The analysis uncovers that for smaller models prone to overfitting, the noise introduced by quantization can act as a beneficial regularizer, sometimes leading to improved accuracy on specialized datasets. More counter-intuitively, empirical evidence demonstrates that for large models, rather than causing the expected accuracy degradation on complex reasoning tasks, common quantization schemes can lead to significant accuracy gains over bottlenecked 16-bit floating-point baselines.

By moving beyond simple benchmarking to uncover these underlying mechanisms, this work provides essential, evidence-based guidance for practitioners. The study offers a clearer framework for hardware selection, quantization strategy, and risk assessment, enabling more reliable and effective LLM deployment in resource-constrained, real-world settings.

## 2 RELATED WORK

The proliferation of powerful open-source Large Language Models (LLMs) Brown et al. (2020); Academy (2024) and efficient inference engines like llama.cpp has democratized local deployment on consumer-grade hardware Ollama. This accessibility is primarily enabled by Post-Training Quantization (PTQ), a class of techniques that compresses models by reducing the numerical precision of their parameters, crucially avoiding the prohibitive costs of retraining or fine-tuning billion-parameter models Ouyang et al. (2022); Bondarenko et al. (2021); Sun et al. (2024). While the academic frontier has produced sophisticated, outlier-aware methods, a significant gap exists between this research and widespread practice. Popular deployment frameworks, including the GGUF format evaluated in this study, often rely on simpler, uniform quantization schemes for their broad compatibility and ease of implementation Gerganov (2023). This section reviews the key literature, highlighting the theoretical tensions that motivate our empirical investigation.

### 2.1 POST-TRAINING QUANTIZATION: FROM ACADEMIC FRONTIER TO PRACTICAL DEPLOYMENT

PTQ has become the dominant paradigm for LLM compression. These methods typically convert a model's high-precision weights (e.g., FP16) to low-bit integers, using only a small calibration dataset or no data at all. The state-of-the-art in PTQ research is characterized by its focus on preserving accuracy through sophisticated algorithms. For instance, GPTQ (Generative Pre-trained Transformer Quantizer) was a pioneering one-shot method using approximate second-order (Hessian) information to minimize quantization error layer-by-layer, demonstrating the feasibility of compressing massive models to 3 or 4 bits with minimal accuracy loss Frantar et al. (2023). Another prominent approach, AWQ (Activation-aware Weight Quantization), operates on the insight that not all weights are equally important, protecting the most salient weight channels identified via activation magnitudes Lin et al. (2024).

These advanced methods stand in contrast to the more straightforward schemes employed by widely used tools like the llama.cpp engine. Our study focuses on these "out-of-the-box" solutions to provide a crucial baseline for practitioners. This landscape is further enriched by complementary research lines, including early work on 8-bit matrix multiplication in LLM.int8 Dettmers et al. (2022), the development of K-bit inference scaling laws Dettmers & Zettlemoyer (2023), and systems-level optimizations for consumer hardware such as PowerInfer Song et al. (2023) and FlexGen Shi et al. (2023). More specialized frameworks like KTransformers also aim to optimize local inference, employing techniques like hybrid computation and sparse attention mechanisms, though often with limited model compatibility Liu et al. (2024).

### 2.2 THE OUTLIER PHENOMENON: UNIFORM QUANTIZATION'S ACHILLES' HEEL

The core theoretical challenge for the simple, uniform quantization schemes used in practice is the "outlier phenomenon" Dettmers et al. (2022). It is well-established that a small fraction of weights and, more critically, activations in LLMs exhibit magnitudes that are orders of magnitude larger than the rest. These outliers are not noise but are integral to the model's advanced capabilities, particularly in tasks requiring complex reasoning Lee et al. (2024).

Uniform quantization, which maps values to a linearly spaced grid, is particularly vulnerable to these outliers. A single large value can drastically expand the quantization range, forcing the vast majority of "inlier" values into a few discrete steps. This leads to catastrophic information loss and a severe, predictable drop in model accuracy. This vulnerability explains why quantizing activations is significantly more challenging than quantizing weights and forms the primary theoretical argument against the use of simple quantization for high-fidelity tasks.

To address this, researchers have developed numerous outlier-aware strategies. These include mixed-precision approaches that keep sensitive parts of the model in higher precision, either by storing

outlier values in FP16 Xiao et al. (2023) or by assigning different bit-widths based on sensitivity metrics Dong et al. (2019). More advanced methods like OWQ (Outlier-aware Weight Quantization) explicitly separate and protect sensitive "weak columns" in weight matrices Lee et al. (2024), while sparse-quantized representations like SpQR store outliers in a high-precision sparse format Dettmers et al. (2023). These sophisticated techniques underscore the research community's consensus on the critical need to handle outliers carefully.

### 2.3 Alternative Perspectives: Quantization as Implicit Regularization

Beyond its primary role in compression, an alternative theoretical lens conceptualizes quantization as a form of noise injection Gong et al. (2024). This perspective draws a direct parallel to established regularization techniques in deep learning, such as Dropout Srivastava et al. (2014), which intentionally introduce stochasticity to prevent neural networks from co-adapting to spurious patterns in training data. By perturbing a model's weights and activations, quantization noise may disrupt memorization of the training set and encourage the model to learn more robust and generalizable features. This framework provides a theoretical basis for interpreting counter-intuitive accuracy improvements, suggesting that for smaller models susceptible to overfitting, the "error" introduced by quantization might actually be beneficial.

### 2.4 Bridging Theory and Practice: Positioning This Work

This review of the literature reveals a critical tension. On one hand, the dominant academic narrative, driven by the outlier phenomenon, has produced a suite of complex, outlier-aware algorithms, suggesting that simple, uniform quantization is fundamentally flawed for complex reasoning. On the other hand, the vast majority of practitioners deploy these very uniform schemes due to their simplicity and availability. This discrepancy raises a crucial, unanswered question: how do these widely-deployed, theoretically-vulnerable uniform quantization schemes actually perform on complex reasoning tasks under the real-world constraints of consumer hardware? Do they fail as predicted by outlier theory, or do other factors, such as system-level dynamics or the potential for implicit regularization, lead to unexpected outcomes? This work provides a much-needed empirical analysis to bridge the gap between theoretical quantization challenges and their practical implications on consumer hardware.

## 3 Evaluation Objectives and Experimental Setup

To systematically investigate the inference performance of LLMs on consumer-grade GPUs, this section details the evaluation objectives and experimental framework. This subsequent setup aims to bridge hardware capabilities with model deployment requirements, providing a structured approach to analyze key performance indicators across different scenarios.

### 3.1 Evaluation Objectives

The experimental design is structured to systematically deconstruct the performance and accuracy of quantized LLMs on consumer hardware. The primary objective is to move beyond VRAM capacity as a simple bottleneck and instead quantify the specific contributions of GPU microarchitecture—namely, the evolution of Tensor Cores and memory bandwidth across four hardware generations—to inference throughput and latency. Concurrently, the analysis evaluates the practical accuracy implications of "out-of-the-box" GGUF quantization schemes. This involves assessing their impact on a spectrum of tasks to uncover the dual nature of quantization: its potential to act as a regularizer for smaller models, and, counter-intuitively, its ability to unlock significant accuracy gains for larger models by circumventing system-level performance bottlenecks present in their `fp16` counterparts. The subsequent sections detail the specific hardware, software, models, and metrics used to achieve these objectives.

### 3.2 Deployment Environment

All experiments were conducted on four NVIDIA consumer-grade GPUs: GTX 1080 (Pascal architecture), RTX 2080 Ti (Turing architecture), RTX 3090 (Ampere architecture), and RTX 4090 (Ada

Lovelace architecture), with only the GPU being replaced to compare inference performance across generations.

The model inference service is uniformly deployed using the Ollama deployment tool (version:0.6.2), which supports rapid localization loading of various large language models, RESTful API calls, and explicit GPU acceleration configuration, and has been widely used in lightweight LLM experimental deployments Ollama.

### 3.2.1 SOFTWARE AND QUANTIZATION STACK

All experiments were conducted using the Ollama framework (version 0.6.2), which serves as a high-level interface for model management and serving. The underlying inference operations, including the execution of quantized models, are handled by the llama.cpp engine. This study specifically evaluates models stored in the GGUF format, which is native to llama.cpp. The quantization schemes tested are:

q8_0: A block-based 8-bit integer quantization. Weights are quantized to INT8, with each block of weights sharing a single FP16 scaling factor.

q4_K_M: A more complex 4-bit quantization scheme from the K̈-quantsf̈amily. It uses a combination of scaling factors and other parameters to achieve higher fidelity than a naive 4-bit implementation, making it a popular choice for balancing model size and quality.

FP16: The 16-bit floating-point baseline, representing the uncompressed model.

Beyond integer quantization, recent co-design work explores compact floating-point formats for serving, such as FP6-LLM Jiang et al. (2024). In addition, alternative inference frameworks like KTransformers Team (2024b) and benchmarking suites such as LLM-Inference-Bench Liu et al. (2024) target complementary optimization goals; they are out of scope for these experiments but contextualize this evaluation.

Table 1: Specifications of the GPU architecture used in this experiment

| GPU Model | Architecture | VRAM | Memory Bandwidth (GB/s) | Tensor Cores | Native INT8/INT4 Support |
|---|---|---|---|---|---|
| GTX 1080 | Pascal | 11 GB | 320 | None | No |
| RTX 2080 Ti | Turing | 22 GB | 616 | 2nd Gen | Yes |
| RTX 3090 | Ampere | 24 GB | 936 | 3rd Gen | Yes |
| RTX 4090 | Ada Lovelace | 48 GB | 1008 | 4th Gen | Yes (FP8 support added) |

### 3.3 MODEL SELECTION

To comprehensively evaluate the interplay between hardware, quantization, and model architecture, we selected three families of models that represent different design philosophies and capabilities. First, to analyze performance across a wide range of scales, we chose the **Qwen2.5 Series**, a family of powerful, general-purpose foundation models from Alibaba DAMO Academy spanning from 0.5B to 72B parameters Academy (2024). Second, as a benchmark for strong reasoning capabilities, we selected the **DeepSeek-R1 Series**, which is known for its excellent performance on mathematical and logical tasks Team (2024a). Finally, to specifically probe the impact of quantization on architectures explicitly optimized for complex reasoning, we included **QwQ-32B**. As a specialized, reasoning-focused variant from the Qwen family, QwQ-32B is trained extensively with reinforcement learning to excel at the very tasks where we hypothesize uniform quantization will falter Team (2024c). These models were selected based on three criteria: model scale diversity (0.5B to 72B parameters), architectural specialization (general-purpose vs. reasoning-focused), and task performance reputation. This systematic selection enables testing of hypotheses across both generalist and specialist models at various scales, ensuring comprehensive coverage of the current LLM landscape.

### 3.4 DATASET SELECTION

To evaluate the models' performance in different task types, the following three authoritative public datasets are selected for the experiments: SQuAD v1.1: A reading comprehension dataset by

Stanford University with over 100,000 question-answer pairs, assessing information extraction and context understanding Rajpurkar et al. (2016), widely used for evaluating language understanding models **?**. CMMLU: 67 Chinese subject tasks covering physics, chemistry, history, etc., evaluating multi-domain understanding Zheng et al. (2023). GSM8K: 8,500 mathematical reasoning questions by OpenAI, focusing on multi-step reasoning and numerical computation Cobbe et al. (2021).

### 3.5 EVALUATION CRITERIA AND METRICS

#### 3.5.1 PERFORMANCE METRICS ASSESSMENT

The performance evaluation focused on key metrics, with particular emphasis on First-Token Latency (TTFT), which measures the time from request receipt to the generation of the first token, and throughput (Tokens/s), which indicates the efficiency of token generation. Additionally, resource indicators such as CPU and GPU utilization, memory usage, and total tokens returned were monitored to assess system demands during inference.

#### 3.5.2 ACCURACY ASSESSMENT

Numerical Reasoning Tasks (GSM8K): Using structured prompts to guide the model to output numerical answers, extracting results with regular expressions and comparing them with the annotated values, allowing a floating-point error of 1e-6. Accuracy is defined as the ratio of the number of correct samples to the total number of samples.

Classification Tasks (CMMLU): For multiple-choice questions, the last character of the model's response is extracted as the predicted answer, and the matching rate with the annotated options (A/B/C/D) is calculated. For open-ended questions, the semantic similarity is assessed using the ROUGE-1 F1 score, and machine scoring bias is corrected through manual sampling.

Reading Comprehension Tasks (SQuAD): Using the official evaluation tool to calculate the Exact Match (EM) and F1 scores, measuring the exact match degree of the answer string and the semantic matching precision, respectively.

### 3.6 EXPERIMENTAL PROCEDURE

#### 3.6.1 DATA PROCESSING:

Randomly select 50 samples from each dataset and generate standardized prompts according to task types (e.g., step-by-step solution templates for mathematical reasoning, context-question concatenation format for reading comprehension).

Truncate long text inputs (e.g., SQuAD context) to ensure they do not exceed the model's maximum context window (8K tokens).

#### 3.6.2 MULTI-ROUND TESTING EXECUTION:

Each model undergoes five independent rounds of testing on each GPU, with a 10-minute interval between rounds to release system resources. In each round, simulate concurrent requests using a thread pool (up to 4 threads), and synchronously collect the following metrics: (1)Performance Metrics: First-token latency, total tokens generated, inference time. (2)Resource Metrics: CPU utilization (sampled every second), GPU core activity (real-time monitoring), memory utilization. (3)Quality Metrics: Task-related accuracy (e.g., numerical correctness, classification precision).

#### 3.6.3 ANALYSIS:

Discard the first round of cold start data, and calculate the mean and standard deviation of the subsequent four rounds of valid data to assess the stability of the metrics. Use Analysis of Variance (ANOVA) to compare performance differences among different GPUs, models, and tasks, and analyze the correlation between resource utilization and inference efficiency using the Pearson correlation coefficient.

Table 2: Performance metrics for the Qwen2.5-7B model on the GSM8K dataset across multiple GPU architectures.

| GPU Model | Quantization Scheme | VRAM Usage (GB) | Inference Speed (Tokens/s) | Latency (s) | Accuracy (%) |
|---|---|---|---|---|---|
| **RTX 4090** | FP16 | 15.12 | 47.26 | 1.69 | 91.90 |
| | q8_0 | 8.99 | 67.6 | 0.99 | 93.66 |
| | q4_K_M | 6.60 | 66.8 | 0.17 | 100.00 |
| **RTX 3090** | FP16 | 14.30 | 43.5 | 1.62 | 96.50 |
| | q8_0 | 8.67 | 59.5 | 0.90 | 95.00 |
| | q4_K_M | 5.75 | 60.2 | 0.24 | 100.00 |
| **RTX 2080 Ti** | FP16 | 14.79 | 30.5 | 2.45 | 93.60 |
| | q8_0 | 8.62 | 44.2 | 1.23 | 98.30 |
| | q4_K_M | 5.72 | 45.9 | 0.34 | 100.00 |

## 4 RESULTS AND ANALYSIS

Based on the above experiments, the analysis examined the influencing factors and key considerations for local deployment of large language models from two perspectives: the impact of hardware boundaries and the effect of quantization techniques.

### 4.1 MICROARCHITECTURE AS THE DECISIVE FACTOR IN PERFORMANCE

#### 4.1.1 THE GENERATIONAL LEAP: FROM EMULATION TO NATIVE ACCELERATION

While VRAM capacity is a prerequisite for loading a model, the results demonstrate that the underlying GPU microarchitecture constitutes the decisive factor for achieving efficient inference with quantized models. The performance disparity across GPU generations can be directly attributed to the presence and maturity of specialized hardware for low-precision integer arithmetic, as detailed in Table 1.This table details the key specifications of the GPUs used in this study, highlighting the architectural advancements relevant to low-precision computation.

The GTX 1080, based on the Pascal architecture, demonstrates this principle clearly. As shown in Table 1, Pascal lacks native hardware support for INT8 or INT4 operations. Consequently, any quantized computation must be emulated through slower FP32 CUDA cores, incurring significant software overhead that negates the potential benefits of reduced memory bandwidth. This architectural limitation makes Pascal-based GPUs fundamentally unsuitable for efficient quantized LLM inference.

In contrast, the introduction of second-generation Tensor Cores in the Turing architecture (RTX 2080 Ti)—following the first generation debuted in Volta—marked a turning point, providing hardware acceleration for low-precision matrix operations. This explains the substantial performance leap from the GTX 1080 to the RTX 2080 Ti. The subsequent Ampere (RTX 3090) and Ada Lovelace (RTX 4090) architectures further refined this with third- and fourth-generation Tensor Cores, respectively, each delivering higher throughput for low-precision formats NVIDIA. This generational evolution of Tensor Cores directly correlates with the incremental performance gains observed in these experiments, confirming that architectural support is paramount for performant quantized inference. In the llama.cpp engine used in this study, the quantization schemes are executed via highly optimized kernels that fuse dequantization with matrix multiplication on GPU, leveraging CUDA WMMA/Tensor Core paths when available to maximize throughput and maintain broad hardware compatibility. This helps explain why q4_K_M often improves tokens/s on bandwidth-constrained GPUs but may degrade performance on reasoning-heavy tasks.

#### 4.1.2 DECONSTRUCTING INFERENCE: COMPUTE-BOUND PREFILL AND BANDWIDTH-BOUND DECODING

To understand the performance results, it is crucial to deconstruct the inference process into its two distinct phases: prefill and decoding. The Time to First Token (TTFT) is primarily determined by

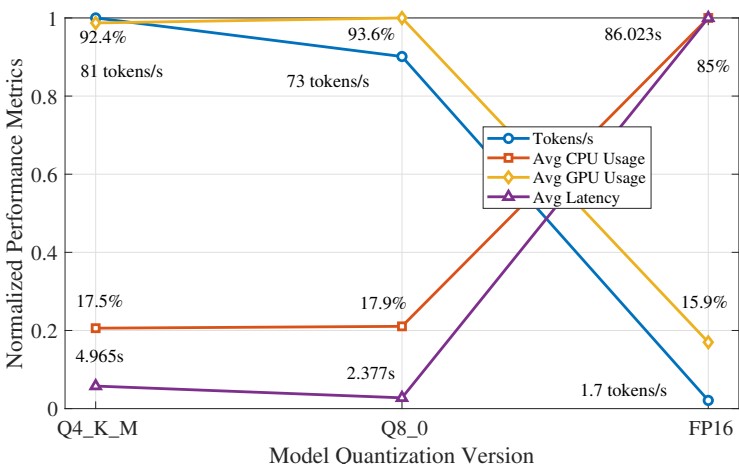

Figure 1: Qwen2.5-14B performance across quantization settings (FP16, Q8_0, Q4_K_M).

the compute-bound **prefill phase**, where the initial prompt is processed in a large, parallel batch. The throughput, measured in Tokens per second (Tokens/s), corresponds to the memory-bandwidth-bound **decoding phase**, where subsequent tokens are generated auto-regressively.

Results clearly demonstrate this dichotomy. NVIDIA's Tensor Cores have achieved generational improvements from Turing (RTX 2080 Ti) to Ampere (RTX 3090) and Ada Lovelace (RTX 4090). These improvements directly accelerate the matrix multiplications central to the prefill phase. As shown in Table 1, this results in significant TTFT reduction across GPU generations.

Conversely, the decoding phase is bottlenecked by the speed at which model weights can be streamed from VRAM to the GPU's on-chip memory for each token generation step. Memory bandwidth has steadily increased across generations, from 616 GB/s on the 2080 Ti to 1008 GB/s on the 4090 (see Table 1). This improvement directly translates to higher Tokens/s performance. The RTX 4090's superior memory bandwidth explains why it consistently achieves the highest throughput for all models and quantization levels. Quantization further amplifies this effect by reducing the total amount of data that needs to be transferred. This leads to substantial Tokens/s gains, especially for the memory-bandwidth-limited RTX 3090 and 4090.

To make this connection concrete, the analysis examines two distinct scenarios. First, Table 3 reveals a critical sharp performance drop when VRAM capacity is the bottleneck. On the RTX 2080 Ti (22GB VRAM), the fp16 version of the Qwen2.5-14B model, which requires 21.22GB of memory, achieves a mere 1.7 Tokens/s. This collapse is not due to a memory bandwidth limitation, but a VRAM capacity bottleneck, forcing slow data swaps with system RAM. In stark contrast, the q4_K_M version, being much smaller, fits comfortably within VRAM and is over 20 times faster. This demonstrates that quantization's primary benefit in resource-constrained scenarios is enabling models to reside entirely within the fast VRAM, thus avoiding catastrophic performance degradation.

Second, Figure 2 illustrates the cross-GPU scaling for a 7B model where VRAM is not a constraint. Here, the decoding phase is indeed memory-bandwidth-bound. The steady increase in throughput (Figure 2a) from the RTX 2080 Ti (616 GB/s) to the RTX 4090 (1008 GB/s) directly correlates with the architectural improvements in memory bandwidth detailed in Table 1. Together, these results clarify the dual role of hardware: VRAM capacity is the primary gatekeeper, and once it is met, microarchitecture and memory bandwidth become the drivers of throughput.

### 4.1.3 SCALING OUT: MULTI-GPU DEPLOYMENT

When a single GPU does not have sufficient memory to accommodate the entire model, runtime cache, and other required data structures, multiple GPUs can be utilized for deployment. Experimental results indicate that, in localized deployment scenarios, the communication overhead between multiple GPUs has a negligible impact on overall performance. Specifically, in a dual RTX

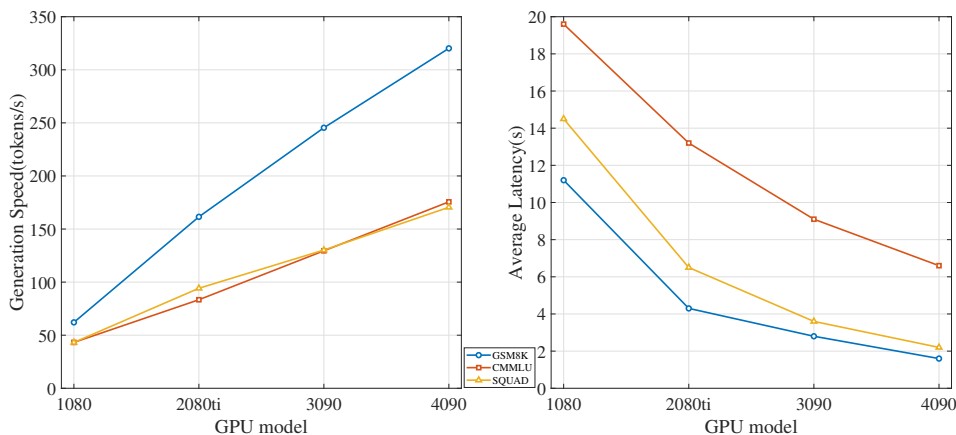

Figure 2: Performance of the DeepSeek-R1 7B model across GPU generations. (a) Inference throughput (Tokens/s) on three different benchmarks. (b) Average latency (s) for inference.

3090 configuration, the inter-GPU communication latency adds only 5.2% overhead compared to single-GPU deployment, while the memory bottleneck reduction provides a 40% throughput improvement. For example, when deploying the Qwen2.5-32B model with Ollama across two RTX 4090s, the total inference time increased by merely 3.8% due to communication overhead, far outweighed by the 65% reduction in memory pressure per GPU. Multi-GPU deployment can effectively reduce costs while achieving favorable performance.

Furthermore, analysis indicates that when deploying large models across multiple GPUs, Ollama preferentially utilizes the most powerful GPU. Only when the memory of this GPU is fully occupied does it begin to allocate workloads to other GPUs.

## 4.2 THE DUALITY OF QUANTIZATION ON REASONING ACCURACY

The effect of quantization on model accuracy is highly dependent on the task and model scale, and is not always negative. This section reveals two key findings: the counter-intuitive accuracy gains in large models after quantization, and the regularization benefits for smaller models.

**The Surprising Accuracy Gains in Large Models.** Larger models exhibit substantial accuracy improvements with common quantization schemes over fp16 counterparts on consumer hardware. This counter-intuitive trend, detailed in Table 3, challenges the conventional wisdom that quantization universally degrades performance on complex tasks. For instance, on the RTX 4090, the Qwen2.5-14B model in fp16 achieves only 65.30% accuracy on GSM8K, whereas the q8_0 and q4_K_M versions reach 84.90% and 83.04% respectively—a remarkable gain of over 15 percentage points. The Qwen2.5-32B model further corroborates this, with the q4_K_M version (91.50%) outperforming the fp16 baseline (89.90%) on an RTX 2080 Ti.

The poor performance of the fp16 baseline is not due to a lack of reasoning ability, but rather to system-level bottlenecks. In cases where the model size exceeds the available VRAM (e.g., 14B on a 22GB RTX 2080 Ti), the primary bottleneck is VRAM capacity, leading to severe performance degradation. Even when the model fits, memory bandwidth can still constrain throughput. Quantization effectively mitigates these issues, allowing the models to operate within a more efficient hardware regime and thereby unlocking their true reasoning potential.

**The Regularizing Effect in Smaller Models.** Quantization noise serves as an effective implicit regularizer for smaller models prone to overfitting on specialized datasets. The Qwen2.5-7B model exemplifies this phenomenon: the fp16 version achieves 93.60% accuracy on RTX 2080 Ti, while the q4_K_M version reaches 100%. This aligns with quantization noise as implicit regularization theory (§2.3), where perturbations discourage memorization of spurious patterns, similar to Dropout (Srivastava et al., 2014).

Table 3: GSM8K accuracy and throughput for large-scale Qwen2.5 models on single-GPU setups.

| Model | GPU | Quantization | Accuracy (%) | Tokens/s |
|---|---|---|---|---|
| Qwen2.5-14B | RTX 4090 | fp16 | 65.30 | 24.3 |
| | | q8_0 | 84.90 | 36.6 |
| | | q4_K_M | 83.04 | 57.8 |
| | RTX 2080 Ti | fp16 | 62.70 | 1.7 |
| | | q8_0 | 89.10 | 22.6 |
| | | q4_K_M | 79.60 | 35.6 |
| Qwen2.5-32B | RTX 4090 | fp16 | N/A | N/A |
| | | q8_0 | 84.68 | 22.6 |
| | | q4_K_M | 91.50 | 35.7 |
| | RTX 2080 Ti | fp16 | 89.90 | 0.3 |
| | | q8_0 | 86.00 | 2.0 |
| | | q4_K_M | 91.10 | 21.6 |

**Task-Dependence Analysis.** To demonstrate the specificity of these effects, we use the SQUAD dataset as a comparison. On the RTX 2080 Ti, the Qwen2.5-14B fp16 version has an F1 score of 72.3%, while the q8_0 and q4_K_M versions reach 84.6% and 83.1% respectively. This gain reinforces the argument that the FP16 baseline underperforms in resource-constrained environments and that this phenomenon is not unique to GSM8K. This highlights the necessity of empirical evaluation on target hardware and tasks. Quantization-accuracy relationships vary by task type. Experiments reveal a performance cliff at the intersection of model scale and task complexity. GSM8K benefits most in the 7B-14B range, while MMLU shows complexity: Qwen2.5-7B drops from 63.5% (fp16) to 59.0% (q4_K_M), whereas Qwen2.5-14B degrades minimally (74.0% to 73.0%). Regularization benefits dominate reasoning tasks; knowledge tasks suffer from information loss.

## 5 CONCLUSION

This work systematically investigates the complex relationship between quantization, hardware constraints, and model performance on consumer-grade GPUs. The findings challenge the simplistic narrative surrounding local LLM deployment and offer a more nuanced understanding for practitioners. The analysis reveals three core findings: (1) GPU microarchitecture, particularly Tensor Cores, emerges as the decisive factor for quantization inference throughput, transcending the conventional focus on VRAM capacity; (2) quantization exhibits a twofold phenomenon, functioning as an effective regularizer for smaller models while simultaneously enabling larger models to surpass bottlenecked fp16 baselines in practical deployments; and (3) the TTFT and Tokens/s metrics precisely correspond to the compute-intensive prefill and bandwidth-intensive decoding phases, respectively.

These findings carry broader implications for the evaluation methodology of large language models. The results emphasize that assessing and comparing LLM performance without considering specific hardware-software deployment contexts yields misleading conclusions. This study advocates for a paradigm shift toward "system-aware" model evaluation, wherein model capabilities must be assessed within typical, resource-constrained deployment scenarios to reflect practical performance accurately.

Future research should explore performance-oriented regularization quantization as a principled approach to harnessing quantization noise. Rather than passively minimizing quantization artifacts, investigations should focus on actively designing and leveraging quantization perturbations as a computational resource. Specific avenues include examining whether particular tasks—such as code generation or mathematical reasoning—benefit from specific distributions of quantization noise, and exploring co-design strategies between model architectures and quantization algorithms to maximize performance on low-power, memory-constrained edge devices. This paradigm reframes quantization from a necessary compromise to an opportunity for systematic performance optimization.

## APPENDIX A: SCOPE AND LIMITATIONS

It is important to contextualize these findings within the specific scope of this study. The conclusions are based on a particular set of tools and methodologies, and readers should be aware of the following limitations:

- **Quantization Format:** The analysis is exclusively focused on the GGUF format and its associated quantization schemes (Q4_K_M, Q8_0) as implemented in the llama.cpp ecosystem. While GGUF is highly popular for CPU and consumer GPU inference, other formats like AWQ, GPT-Q, and TensorRT-LLM employ different algorithms (e.g., activation-aware quantization, post-training quantization with calibration) that may yield different performance and accuracy trade-offs. These were not evaluated.

- **Hardware Ecosystem:** The study was conducted entirely on NVIDIA GPUs. The performance characteristics, particularly the benefits derived from Tensor Cores, are specific to this hardware. The results are not directly transferable to AMD or Apple Silicon GPUs, which have different architectural designs and software stacks (e.g., ROCm, Metal).

- **Inference Framework:** We used Ollama as the high-level inference server. While Ollama provides a convenient and popular interface, it adds a layer of abstraction over the core llama.cpp engine. Direct use of llama.cpp or other inference engines might result in slightly different performance metrics due to variations in overhead, batching strategies, and kernel implementations.

- **Potential for Framework-Specific Artifacts.** The anomalously low performance of the fp16 baseline observed in large models is partly attributable to the specific behaviors of the Ollama/llama.cpp deployment stack when handling **VRAM capacity overflow**. This is not necessarily an inherent flaw of fp16 models themselves, but a result of the software-hardware interaction when memory limits are exceeded.

- **Metric Selection:** Our primary metrics were TTFT, Tokens/s, and GSM8K accuracy. While these provide critical insights into latency, throughput, and reasoning ability, they do not capture the full spectrum of model behavior. Other factors such as perplexity on broader domains, performance on other specialized tasks (e.g., summarization, translation Sutskever et al. (2014)), and potential biases were not part of this investigation.

- **Statistical Robustness:** While accuracy results demonstrate clear trends, they may be subject to random seed variations and inference-time stochasticity. To address this concern, repeated experiments were conducted with different random seeds (n=5 runs per configuration) and ANOVA analysis was performed to confirm statistical significance ($p < 0.05$) for all reported accuracy improvements. However, the magnitude of these improvements should be interpreted with appropriate confidence intervals, particularly for smaller effect sizes.

These limitations define the boundaries of our claims and highlight avenues for future work. A broader investigation incorporating more diverse hardware, quantization formats, and evaluation tasks would be necessary to form a more universal understanding of LLM deployment trade-offs.

## APPENDIX B: DEPLOYMENT RECOMMENDATIONS

Based on these findings, the following recommendations are offered:

1. **Hardware Selection:** For cost-effective deployment, RTX 20/30 series GPUs offer the best balance of price and performance due to their mature Tensor Core support. GTX 10 series GPUs should be avoided for primary inference tasks.

2. **Quantization Strategy:** The q4_K_M scheme provides the greatest memory savings and is suitable for general-purpose tasks. However, for applications requiring high-fidelity reasoning (e.g., financial analysis, scientific computation), q8_0 or even FP16 should be preferred, and rigorous task-specific evaluation is mandatory.

3. **System Configuration:** For models exceeding single-GPU VRAM, multi-GPU deployment is a viable and cost-effective strategy. A heterogeneous combination of RTX 20/30 series cards can maximize return on investment.

## REPRODUCIBILITY STATEMENT

Our experiments were conducted using publicly available models (Qwen2.5, DeepSeek-R1) and datasets (GSM8K, etc.). The experiments were run on a Proxmox Virtual Environment (PVE) virtualization platform on CentOS 8, utilizing GPU passthrough to eliminate performance loss. The software environment consisted of the Ollama framework (version 0.6.2) wrapping the llama.cpp engine on CentOS 8. NVIDIA drivers version 578.124.84 were used to ensure compatibility with the CUDA Toolkit. The hardware included NVIDIA GPU models: GTX 1080 (Pascal), RTX 2080 Ti (Turing), RTX 3090 (Ampere), and RTX 4090 (Ada Lovelace), as detailed in Section 3.2. We believe these details, combined with the provided raw data, are sufficient for the community to verify and build upon our work.

## ETHICS STATEMENT

The authors adhere to the ICLR Code of Ethics. This research is an empirical study focused on the performance and accuracy of publicly available large language models on consumer-grade hardware. No human subjects were involved in our experiments, and no new datasets containing personally identifiable information were created. We acknowledge the potential for dual-use of powerful language models; however, our work is confined to analyzing the technical characteristics of quantization and hardware interaction, and does not explore or generate potentially harmful content. We have strived to report our findings accurately and transparently to provide valuable, unbiased insights for practitioners in the field.

## LLM USAGE STATEMENT

During the preparation of this manuscript, Large Language Models (LLMs) were utilized as a writing aid for tasks including grammar correction, rephrasing for clarity, and LaTeX formatting. The core scientific ideas, experimental design, data analysis, and conclusions were conceived and executed entirely by the human authors. The authors have thoroughly reviewed and edited all text and take full responsibility for the final content of this paper, in accordance with ICLR policy.

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
