# OpenReview forum: "Microarchitecture Is Destiny: Performance and Accuracy of Quantized LLMs on Consumer Hardware"
_ICLR.cc/2026/Conference — ICLR 2026 Conference Desk Rejected Submission_

### Official Review · Reviewer_ERPp · 2025-10-24

**Soundness:** 2
**Presentation:** 2
**Contribution:** 2
**Rating:** 4
**Confidence:** 3

**Summary:**

This paper empirically studies how quantized large language models (LLMs) perform across multiple GPU generations. The authors evaluate four NVIDIA architectures (Pascal, Turing, Ampere, and Ada Lovelace) using three model families (Qwen 2.5, DeepSeek-R1, QwQ-32B) and three datasets (GSM8K, SQuAD, CMMLU).
They report three main findings: (1) once VRAM capacity suffices to load a model, inference throughput is primarily determined by Tensor Core generation and memory bandwidth; (2) quantization can improve accuracy for large models by mitigating system-level bottlenecks; and (3) quantization can behave as an implicit regularizer for smaller models, improving generalization.

**Strengths:**

- The experiments span four GPU generations, multiple model sizes, and diverse benchmarks, offering strong empirical coverage.

- The compute-bound versus bandwidth-bound distinction provides clear insight into performance bottlenecks during inference.

**Weaknesses:**

- The result that VRAM capacity acts as a bottleneck is not novel. It is already well-understood that once a model fits in memory, inference becomes bandwidth-bound.
- The paper dedicates significant space to infrastructure and setup details, but lacks deeper theoretical grounding to explain the observed behaviors (e.g., why quantization sometimes improves accuracy).
- Only q4_K_M and q8_0 quantization schemes are tested. Excluding other widely used techniques such as GPTQ, AWQ, or SmoothQuant limits the generalizability of the conclusions.
- Some experimental results (e.g., FP16 accuracy degradation) require further analysis to isolate whether they stem from framework overhead (Ollama/llama.cpp) or hardware-level constraints.
- Evaluation on older GPUs (Pascal, Turing) offers limited relevance for modern LLM deployment, as these architectures lack key optimizations such as advanced Tensor Cores and efficient memory pipelines.
- Minor: missing citation on line 218. Kindly proof-read the paper thoroughly.

**Questions:**

- Can the FP16 accuracy drop be reproduced using native llama.cpp or TensorRT-LLM to rule out framework-induced effects?
- How consistent are the accuracy improvements across random seeds and datasets?
- Would including modern quantization algorithms (e.g., GPTQ or AWQ) alter the key trends?

---

### Official Review · Reviewer_Lfti · 2025-10-26

**Soundness:** 1
**Presentation:** 1
**Contribution:** 1
**Rating:** 2
**Confidence:** 5

**Summary:**

The paper evaluates the accuracy and throughput impacts of quantization on consumer-grade NVIDIA GPUs across multiple generations. Authors argue that, for both small and large models, quantization noise can act as a form of regularization, yielding 4-bit and 8-bit performance that sometimes surpasses the FP16 baseline.

**Strengths:**

* Measures both accuracy and throughput across multiple generations of NVIDIA consumer GPUs, providing a cross-hardware view of quantized LLM inference.

**Weaknesses:**

* Questionable evaluation outcome. Table 3 reports 8-bit/4-bit models outperforming the FP16 baseline by large margins. Attributing >~20% accuracy gains to “regularization” from quantization is not credible. In practice, any quantization-induced improvements over FP16 are typically tiny (≈≤0.5%) and task-dependent.
* Missing evidence for Section 4.1.3. The section references results that are not shown. A corresponding table/figure is needed to interpret the claims and their context.
* Minor: L218 missing citation.
* Minor: L349 likely refers to Table 2, not Table 1.

**Questions:**

See weakness

---

### Official Review · Reviewer_qr1a · 2025-10-30

**Soundness:** 1
**Presentation:** 1
**Contribution:** 1
**Rating:** 0
**Confidence:** 5

**Summary:**

This paper analyzes the impact of quantization on large language models (LLMs) in terms of accuracy and inference speed across various GPUs.

**Strengths:**

No clear strengths are identified in this paper.

**Weaknesses:**

- While the evaluation setup (Section 3.3 and 3.4) describes various types of LLMs and tasks, the analysis is conducted only on a single LLM (Qwen2.5) and a single dataset (GSM8K), which limits the generality and robustness of the conclusions.
- The findings mostly restate well-known characteristics of quantization. For example, that quantization can occasionally improve accuracy when a model is excessively large relative to the simplicity of the target task.
- When the model fits entirely within GPU memory, it is straightforward that GPUs supporting lower-precision computation will deliver better throughput. Thus, the conclusions presented in this paper lack new insight.

**Questions:**

Please check the weakness

---

### Official Review · Reviewer_eNtZ · 2025-11-01

**Soundness:** 1
**Presentation:** 1
**Contribution:** 1
**Rating:** 2
**Confidence:** 3

**Summary:**

This paper conducts a system-aware empirical study of two LLMs (DeepSeek-R1 and Qwen2.5) across four consumer-grade NVIDIA GPUs (1080/2080Ti/3090/4090). The authors evaluate multiple quantization methods (fp16/q8/q4) on 3 datasets (GSM8k/CMMLU/SQuAD) and surprisingly find that low-bit quantization schemes have accuracy improvements over fp16. They hypothesize that this is due to quantization working as implicit regularization for smaller models.

**Strengths:**

The experiments conducted in this paper are very close to the real use case of end-users (ollama framework and consumer-grade GPUs).

**Weaknesses:**

- The experiment is conducted on limited datasets (3 datasets) with limited metrics (accuracy).
- There might be flaws during their experiments. For example, in Table 2, the accuracy of FP16 is different on multiple GPUs. The 100% accuracy on the Q4 scheme also looks suspicious.
- The authors do not provide a convincing explanation of how the accuracy improvement is related to implicit regularization.

**Questions:**

- Please provide a detailed analysis of the experiment results. For example, in Table 2, why does fp16 have different accuracy on different GPUs? Are the experiment setups aligned?
- Please extend the experiments to more datasets and more metrics to reproduce the observation of accuracy gain.

---

### Note · Program_Chairs · 2026-01-17
**Submission Desk Rejected by Program Chairs**

The following references in this submission do not refer to real documents and/or have major errors in bibliographic information:

 Zhihang Sun, Zhiqiang Liu, Zhiyu Wang, and Jinyang Li. A comprehensive study on activating and weight quantization for large language models. arXiv preprint arXiv:2402.18158, 2024.